# Pushing the Limits of Sparsity: A Bag of Tricks for Extreme Pruning

**Andy Li**  *a.li.21@abdn.ac.uk*
*Department of Computing Science*
*University of Aberdeen, UK*

**Aiden Durrant**  *aiden.durrant@abdn.ac.uk*
*Department of Computing Science*
*University of Aberdeen, UK*

**Milan Markovic**  *milan.markovic@abdn.ac.uk*
*Department of Computing Science & Interdisciplinary Institute*
*University of Aberdeen, UK*

**Tianjin Huang**  *t.huang2@exeter.ac.uk*
*Department of Computer Science*
*University of Exeter, UK*

**Souvik Kundu**  *souvikk.kundu@intel.com*
*Intel Labs, USA*

**Tianlong Chen**  *tianlong@cs.unc.edu*
*Department of Computer Science*
*University of North Carolina at Chapel Hill, US*

**Lu Yin**  *l.yin@surrey.ac.uk*
*School of Computer Science and Electronic Engineering*
*University of Surrey, UK*

**Georgios Leontidis**  *georgios.leontidis@abdn.ac.uk*
*Department of Computing Science & Interdisciplinary Institute*
*University of Aberdeen, UK*

**Reviewed on OpenReview:** *https://openreview.net/forum?id=XX9JdOJD8R*

## Abstract

Pruning of deep neural networks has been an effective technique for reducing model size while preserving most of the performance of dense networks, crucial for deploying models on memory and power-constrained devices. While recent sparse learning methods have shown promising performance up to moderate sparsity levels such as 95% and 98%, accuracy quickly deteriorates when pushing sparsities to extreme levels due to unique challenges such as fragile gradient flow. In this work, we explore network performance beyond the commonly studied sparsities, and develop techniques that encourage stable training without accuracy collapse even at extreme sparsities, including 99.90%, 99.95% and 99.99% on ResNet architectures. We propose three complementary techniques that enhance sparse training through different mechanisms: 1) Dynamic ReLU phasing, where DyReLU initially allows for richer parameter exploration before being gradually replaced by standard ReLU, 2) weight sharing which reuses parameters within a residual layer while maintaining the same number of learnable parameters, and 3) cyclic sparsity, where both sparsity levels and sparsity patterns evolve dynamically throughout training to better encourage parameter exploration. We eval-

uate our method, which we term **E**xtreme **A**daptive **S**parse **T**raining (EAST) at extreme sparsities using ResNet-34 and ResNet-50 on CIFAR-10, CIFAR-100, and ImageNet, achieving competitive or improved performance compared to existing methods, with notable gains at extreme sparsity levels. Code is available at `https://github.com/TensorStrike/ES2`.

# 1 Introduction

Network pruning (Han et al., 2015a;b; LeCun et al., 1990; Liu et al., 2017; Li et al., 2016; Kusupati et al., 2020) is a widely-used technique for reducing a network's parameters and compressing its size. Reducing model sizes is crucial for deploying models on edge devices with limited resources. Conventionally, pruning methods have focused on reducing parameters from pre-trained models. However, it requires at least as much computation as training a dense model as it must converge before pruning takes place. The Lottery Ticket Hypothesis work (Frankle & Carbin, 2018; Frankle et al., 2020a; Malach et al., 2020) gives theoretical foundation that subnetworks have the potential to reach full performance even when trained from an initially sparse state. This insight has recently gained much traction in sparse training, a paradigm in which sparse networks are trained from scratch without the need for dense pre-training.

Sparse training methods can be broadly classified into two categories: static sparse training (SST), also sometimes referred to as Pruning at Initialization (PaI), where the sparsity pattern is pre-determined at initialization and remains fixed throughout training (Lee et al., 2019; Wang et al., 2020; Tanaka et al., 2020a; De Jorge et al., 2020), and dynamic sparse training (DST), where the sparsity pattern continuously evolves during training (Mocanu et al., 2018; Dettmers & Zettlemoyer, 2019; Evci et al., 2021; Yin et al., 2022). These methods perform well at moderate sparsity commonly up to 98%, but experience rapid accuracy degradation beyond this due to limitations such as layer collapse and gradient issues (Tanaka et al., 2020b; Wang et al., 2020). This raises a fundamental question - what are the limits of sparse training? Understanding these boundaries is both scientifically valuable for characterizing sparse learning dynamics and potentially relevant for future deployment scenarios, where resource constraints continue to be problematic. However, extreme sparsities remain relatively understudied, with only a few methods attempting to maintain usable accuracy at these extreme levels (Tanaka et al., 2020b; De Jorge et al., 2020; Price & Tanner, 2021).

Our work proposes a collection of adaptive methods that can achieve meaningful performance at extreme sparsities. By leveraging three core strategies - 1) phased Dynamic ReLU activation, 2) weight sharing, 3) cyclic sparsity scheduling - we present a flexible framework that is capable of optimizing performance at extreme sparsities. Each component independently enhances the sparse model's capacity as seen in our ablation studies, where individual strategy improves performance on their own. These methods can also cohesively achieve an even stronger performance when combined. This framework, which we term **E**xtreme **A**daptive **S**parse **T**raining (EAST), offers a modular approach that can be adapted to existing DST frameworks, postponing their total performance collapse. Through empirical analysis, we find that EAST prevents gradient vanishing and encourages parameter exploration, achieving notable performance gain over our baselines. Our primary focus is on the final compressed model for deployment rather than training efficiency, as these represent distinct optimization objectives in resource-constrained scenarios. Our contribution and key achievements can be summarized as follows:

- We introduce EAST - a method that combines phased Dynamic ReLU activation, weight sharing, and cyclic sparsity scheduling, aimed at retaining performance past common sparsity thresholds.

- We perform extensive empirical evaluation of EAST on image classification tasks, and demonstrate competitive performance at sparsities beyond 99.90%.

- We explore and demonstrate that DyReLU when used as a drop-in replacement for ReLU at initialization and then replaced completely can still improve the parameter exploration at extreme sparsities, leading to higher performance than sparse models trained using ReLU only.

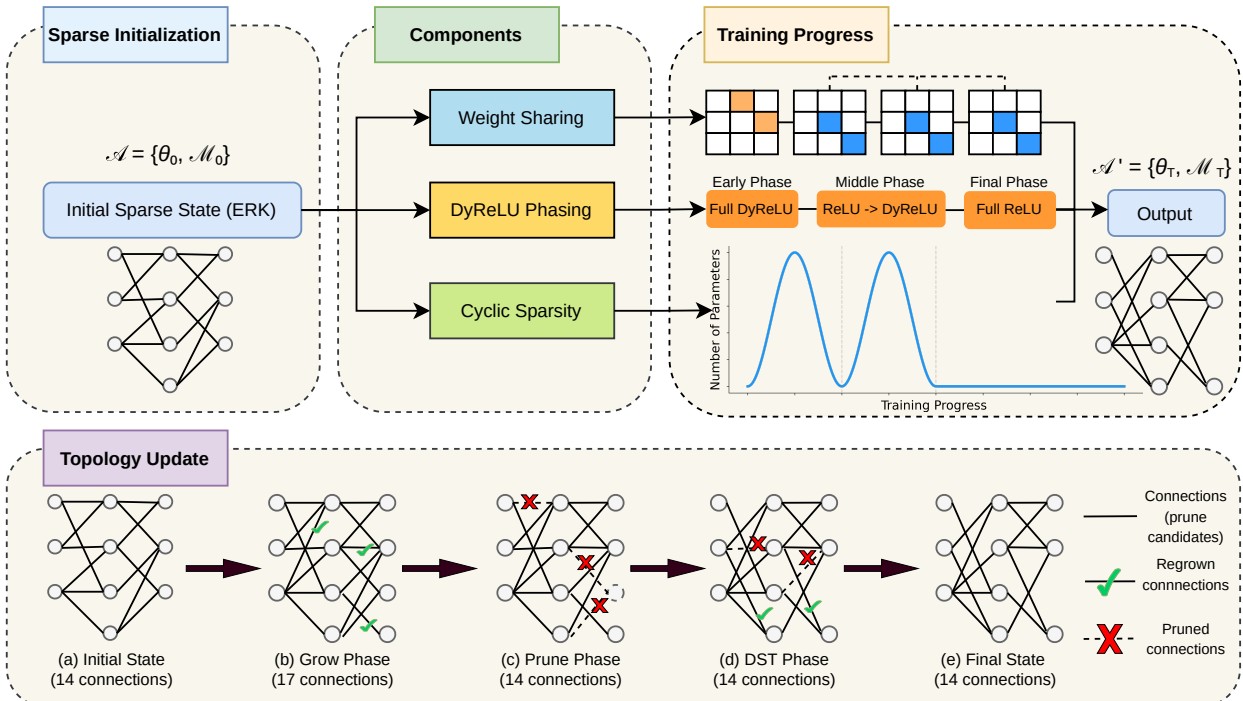

Figure 1: Illustration of EAST (top). Starting with ERK-initialized sparse network $\mathcal{A} = \{\theta_0, \mathcal{M}_0\}$ at sparsity s, EAST employs three key components: DyReLU phasing, weight sharing, and cyclic sparsity to transform the network to final state $\mathcal{A}' = \{\theta_T, \mathcal{M}_T\}$, achieving meaningful performance at extreme sparsity levels. The topology update box (bottom) illustrates the connectivity change throughout training: connections are first grown, then pruned, and eventually maintained with a fixed sparsity update schedule until completion.

- Unlike existing DST methods that maintain fixed sparsity levels, EAST introduces cyclic sparsity scheduling where both patterns and sparsity levels evolve dynamically during training. Our cyclic sparsity schedule allows models to explore parameters and stabilize at extreme sparsity, as evidenced by our ablation studies.

- Unlike DCTpS, EAST achieves extreme sparsity without requiring dense matrix computations for inference, resulting in a truly lightweight and efficient model for edge deployment. We also demonstrate the versatility of our method by combining with DCT layers on top of DST, consistently improving accuracy at the tested settings.

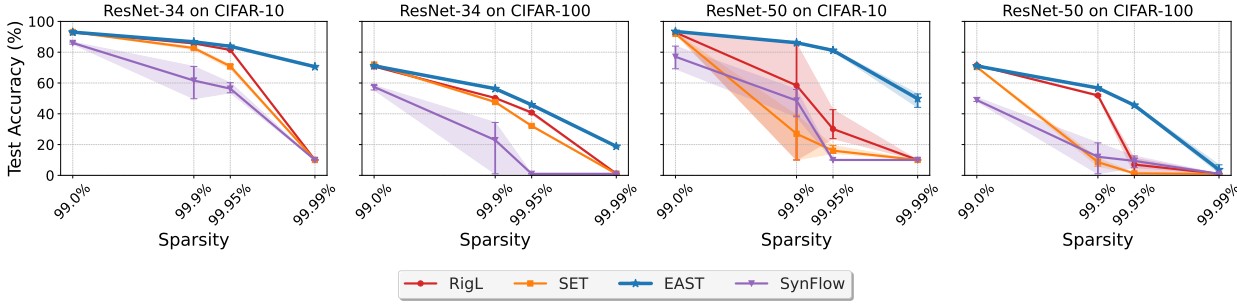

Figure 2: Comparison of test accuracies across different sparsities. Each point represents median accuracy over three runs with different seeds, and the shaded regions highlight the variability across runs.

## 2 Related Work

**Dynamic Sparse Training.** Early work by Mocanu et al. (2018) demonstrated the viability of DST through SET, which used magnitude pruning and random weight regrowth. Subsequent criteria for regrowing connections have been developed, such as using the momentum of parameters in SNFS Dettmers & Zettlemoyer (2019) or their gradient in RigL Evci et al. (2021) to determine the salience of deactivated connections. GraNet Liu et al. (2021a) adapted RigL's approach by starting with half the parameters and gradually increasing to target sparsity instead of starting from the target sparsity. ITOP Liu et al. (2021b) provides a deeper understanding of pre-existing DST methods and enhances parameter exploration during sparse training. Supticket Yin et al. (2022) further enhances the performance of DST by leveraging weight averaging at the late training stage. NeurRev Li et al. (2024) addressed the issue of dormant neurons by removing harmful negative weights. Recent advances include Top-KAST Jayakumar et al. (2020), which maintains an auxiliary set of weights for enhanced exploration, and BiDST Ji et al. (2024) which formulates DST as a bi-level optimization problem for joint weight-mask optimization. MEST Yuan et al. (2021) introduces a memory-efficient sparse training framework for edge devices, while Chase Yin et al. (2024) adapts dynamic unstructured sparsity into hardware-friendly channel-wise sparsity, both optimizing performance on resource-constrained systems without sacrificing accuracy. SRigL Lasby et al. (2024) extends the RigL-based pipeline to structured sparsity and imposes a constant fan-in N:M for better hardware efficiency. CHT Zhang et al. (2025) adopts a regrowth strategy that does not require the use of gradient information in backpropagation, and solely relies on topological information instead, allowing the regrowth computation to be cheaper. DSCR Wu et al. (2025) shows that, at moderate sparsity, dynamically sparse models can outperform dense ones in robustness to common image corruptions. DynaDiag Wu et al. (2025) restricts parameters to a diagonal sparsity pattern, which preserves full input–output coverage while enabling GPU-friendly structured sparsity and dynamic mask updates within that pattern.

**Extreme Sparsity Methods.** Most current methods focus on sparsity levels at which pruned networks match or closely approximate the performance of their dense counterparts, a concept referred to as *matching sparsity* (Frankle et al., 2020b). However, there is growing interest in extreme sparsities, where accuracy trade-offs become inevitable but potentially worthwhile for substantial computational gains. Yet there remain challenges, such as maintaining stable gradient flow and preventing layer collapse, which conventional pruning methods struggle to address. GraSP Wang et al. (2020) and SynFlow Tanaka et al. (2020b) specifically addressed the gradient preservation and layer collapse issues at high sparsities via careful network initialization. SNIP Lee et al. (2019) and SNIP-it Verdenius et al. (2020) proposed PaI methods that showed promising results at higher sparsities, though primarily focused on static masks. De Jorge et al. (2020) demonstrated that methods like SNIP and GraSP can perform worse than random pruning at 99% sparsity and beyond, and proposed iterative pruning approaches (Iter-SNIP and FORCE) that maintained meaningful performance up to 99.5% sparsity through gradual parameter elimination. DCTpS Price & Tanner (2021) introduced "DCT plus Sparse" layers that combined a fixed dense DCT offset matrix with trainable sparse parameters to maintain propagation at extreme sparsities up to 99.99%.

**Activation Functions.** Activation functions in sparse networks have traditionally relied on static activation functions such as ReLU (Nair & Hinton, 2010), which applies a fixed threshold to the input. Parametric variants such as PReLU He et al. (2015) introduced learnable parameters to adapt the negative slope, offering slight improvements in gradient flow. Swish Ramachandran et al. (2017) and Mish Misra (2019) aim to smooth gradients for better optimization but remain static throughout training. Dynamic ReLU (DyReLU) Chen et al. (2020) introduced an adaptive activation that dynamically adjusts the slopes of a ReLU based on the input.

## 3 Methodology

### 3.1 Activation Function

In this work, when initializing a network, we replace the standard ReLU activation with a non-ReLU activation. We tested several activations and found that DyReLU performed best, specifically the DyReLU-B variant, which dynamically adapts and adjusts channel-wise activation coefficients during training, enabling

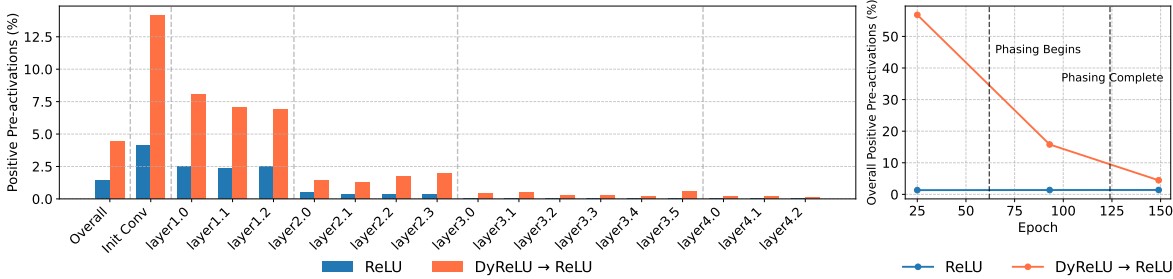

Figure 3: Positive pre-activations analysis in ResNet-34 at 99.99% sparsity. The left figure shows layerwise-comparison of positive pre-activations after DyReLU is completely converted to ReLU. The right figure shows their overall amount before, during and after DyReLU phasing.

richer expressivity during early training. As training progresses, we gradually transition from DyReLU to standard ReLU, eliminating the additional parameters. Preliminary details about DST, DyReLU and details on phasing out DyReLU can be found in the Supplementary Materials.

**Understanding Gradient Flow Through Neuron Activity.** With ReLU, non-positive pre-activations would result in zero gradient and prevent the learning signal from propagating. To better understand why DyReLU phasing improves performance at extreme sparsity, we analyzed pre-activation values throughout training. As shown in Figure 3, the model trained solely with ReLU maintains around 1.3% positive pre-activations throughout training, with many deeper layers showing none, interrupting gradient propagation.

In contrast, our DyReLU phasing approach naturally begins from a high 56.8%, decreases during the transition phase, and stabilizes at 4.5% even after DyReLU is completely converted to ReLU, three times higher than with ReLU alone. The deeper layers also retain some positive pre-activations, preserving critical activation pathways that would otherwise be lost. This allows extremely sparse networks to continue learning where conventional approaches would struggle due to interrupted gradient propagation. We detail the ablation studies on the effect of DyReLU phasing in Section 4.2.

This mechanism of preserving activation pathways directly translates to sustained gradient flow during training. As demonstrated in Figure 5, networks trained with DyReLU phasing maintain healthy gradient norms even after complete conversion to standard ReLU, whereas networks trained solely with ReLU exhibit near-zero gradient flow and complete performance collapse.

**Phasing Out DyReLU.** As training progresses, we phase out all DyReLU and gradually replace them with the standard ReLU. We introduce a hyperparameter that regulates the contribution of DyReLU during training. It makes full contribution when it is 1, and gradually decreases to 0 as training progresses, eventually reaching 0 before the first learning rate decay, at which point DyReLU has been completely replaced by ReLU. We find that this helps the network settle into a more stable configuration, as DyReLU is no longer needed once the network has explored the parameter space adequately. This process removes all DyReLU-introduced parameters.

The *decay factor* for this transition is denoted as $\beta(t)$, which decays linearly over the course of training:

$$\beta(t) = 1 - \frac{t - t_{\text{start}}}{t_{\text{end}} - t_{\text{start}}}$$

where $t_{\text{start}}$ and $t_{\text{end}}$ denote the start and end epochs of DyReLU phasing. We choose $t_{end}$ to be the epoch before the first learning rate schedule update in all of our runs. We initialize with:

$$f(x; t = 0) = f_{DyReLU}(x)$$

During the transition, the activation function progressively shifts toward ReLU:

$$f(x; t) = \beta(t) \cdot f_{DyReLU}(x) + (1 - \beta(t)) \cdot f_{ReLU}(x)$$

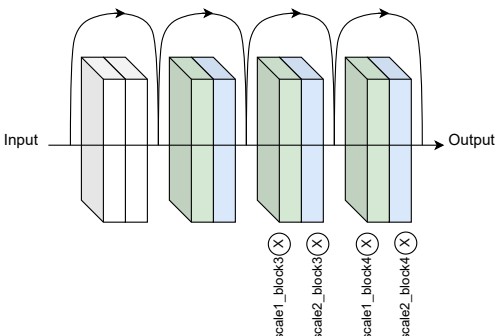

Figure 4: A layer in ResNet-34 with 4 blocks. Block 3 and Block 4 share parameters (and the masks) with block 2. Specifically, conv1 layer (green) of block 3 reuses conv1 layer of block 2, and is multiplied by a learnable scaling factor; similarly, its conv2 layer (blue) reuses conv2 layer of block 2, and is multiplied by another scaling factor.

By the end of phasing $t = T_{\text{end}}$, we have:

$$f(x; t = T_{\text{end}}) = f_{ReLU}(x)$$

This smooth transition from a dynamic to static activation function prevents the network from overfitting to a complex activation pattern while still reaping the benefits of DyReLU in the early stages of training.

## 3.2 Weight Sharing

Much research in sparse training has been about mask optimization - determining the best subset of parameters to keep. However, at extreme sparsities, gradient flow can collapse even with an optimal parameter subset. Therefore, we ask a different question: *Can we increase the number of parameter paths in the computational graph while maintaining the same number of learnable parameters?* Our approach creates multiple references to the same parameter tensors, effectively allowing one learnable parameter to contribute to multiple locations in the forward/backward pass without additional storage cost.

Our weight sharing mechanism strategically reuses parameters within each layer's residual blocks, allowing the remaining active parameters to participate in multiple paths of the computational graph. If, at a certain sparsity, we have $n$ remaining parameters in the original model, we keep exactly this many unique parameters to learn, but some of them now appear in multiple locations in the computational graph, multiplying their contributions without increasing storage requirements. During backpropagation, gradients from multiple blocks sharing the same parameters are automatically summed by PyTorch's autograd, strenghening the learning signal for these weights. Each sharing block includes learnable scaling factors that allow the blocks to specialize while maintaining parameter efficiency.

Specifically, we choose a block within a layer and share its parameters (along with their masks) with all blocks that follow it. These later blocks reference the single block tensor that they share with, and multiply by a learnable scaling factor for forward and backward passes. For example, in a ResNet layer with 4 blocks (Figure 4), if the second block shares parameters with block 3 and 4, a parameter in block 2 effectively appears three times in the computation graph while only being stored and optimized once.

Since multiple computational paths now reference the same learnable parameters, we adjust our sparsity calculation to reflect the true number of unique learnable parameters rather than the apparent network size. Thus, sparsity $s = 1 - \frac{\|\mathcal{M} \odot \theta_{\text{learnable}}\|_0}{\|\theta\|_0}$, where $\theta_{\text{learnable}}$ represents only the unique learnable parameter count being optimized. $\theta$ is the theoretical parameter count in the original network with a mask $\mathcal{M} \in \{0, 1\}^{|\theta|}$. Formally, let $L$ be the number of residual blocks in a layer, and $R$ be the block that shares parameters ($1 < R \leq L$), where the remaining $L - R$ blocks share parameters with the $R$-th block. Let $\theta_i$ represent the

parameters of the $i$-th block. Our weight sharing scheme can be expressed as:

$$\theta_i = \begin{cases} \theta_i & \text{if } i \leq R \\ \theta_R & \text{if } R < i \leq L \end{cases}$$

### 3.3 Cyclic Sparsity

While classic DST methods such as SET and RigL adjust the sparsity pattern, sparsity $s$ remains constant throughout training - pruning and regrowing of weights take place at the same rate. i.e., $\mathcal{M}(t) = \text{Prune}(\theta(t)) \cup \text{Grow}(\nabla_\theta \mathcal{L})$. In our work, we introduce a cyclic sparsity schedule where the network's sparsity level itself evolves dynamically during training. Instead of maintaining constant sparsity, we cyclically adjust the sparsity $s(t)$ between a maximum value $s_{\max}$ and a minimum value $s_{\min}$ over a period $T_c$. This creates phases of higher and lower connectivity, allowing the network to explore parameters more effectively, preventing the gradient issues that plague static extreme sparsity training. Once the cyclic phase ends at $T_c$, we fix the sparsity at $s_{\max}$ and switch to standard DST updates. Details of the method are summarized in Algorithm 1.

---

**Algorithm 1** Pseudocode for EAST

**Input:** Sparse neural network with weights $\theta$, maximum sparsity $s_{\max}$, minimum sparsity $s_{\min}$, end of cyclic sparsity $T_c$, update frequency $\Delta T$, prune rate $r_p$, DyReLU phase start $T_s$, end $T_e$
**Output:** Trained $\theta_s$
1: Initialize: $\theta_s$ at $s_{\max}$, $\phi \leftarrow \text{DyReLU}$
2: **for** $t = 1$ to $T_{\text{end}}$ **do**
3:      **if** $T_s \leq t \leq T_e$ **then**                                                     ▷ DyReLU to ReLU
4:          $\phi \leftarrow \beta \text{DyReLU} + (1 - \beta) \text{ReLU}$
5:          $\text{Update}(\beta)$
6:      **end if**
7:      **if** $t \bmod \Delta T == 0$ **then**
8:          **if** $t <= T_c$ **then**
9:              $s_{\text{target}} \leftarrow \text{CyclicSchedule}(t, s_{\min}, s_{\max}, T_c)$                  ▷ Update target sparsity
10:              **if** $s_{\text{target}} > s_{\text{current}}$ **then**
11:                  $\theta_s \leftarrow \theta_s - \text{ArgTopK}(-|\theta_s|, (s_{\text{target}} - s_{\text{current}})\|\theta_s\|_0)$         ▷ Magnitude pruning
12:              **else if** $s_{\text{target}} < s_{\text{current}}$ **then**
13:                  $\theta_s \leftarrow \theta_s + \text{ArgTopK}(|\nabla_\theta \mathcal{L}|, (s_{\text{current}} - s_{\text{target}})\|\theta_s\|_0)$           ▷ Gradient regrowth
14:              **end if**
15:          **else**                                                   ▷ Switch to fixed update
16:              $\theta_s \leftarrow \theta_s - \text{ArgTopK}(-|\theta_s|, r_p\|\theta_s\|_0)$
17:              $\theta_s \leftarrow \theta_s + \text{ArgTopK}(|\nabla_\theta \mathcal{L}|, r_p\|\theta_s\|_0)$
18:          **end if**
19:      **end if**
20: **end for**
21: **return** $\theta_s$

---

## 4 Experiments

**Evaluation Protocol.** Our experiments include image classification using ResNet-34 and ResNet-50 He et al. (2016) on the benchmark datasets CIFAR-10 and CIFAR-100, as well as ResNet-50 on ImageNet, following the same evaluation protocol as other SOTA methods in terms of datasets and backbones. We compare our method with DST methods SET Mocanu et al. (2018) and RigL Evci et al. (2021), and a competitive SST method SynFlow Tanaka et al. (2020b), which also experiments at very high sparsities. For SynFlow, we use the official repository with its default hyperparameters to evaluate its performance. For SET and RigL, we use the repository provided by ITOP Liu et al. (2021b). We keep all hyperparameters as

| Method | CIFAR-10 | | | | CIFAR-100 | | | |
|---|---|---|---|---|---|---|---|---|
| | 99% | 99.90% | 99.95% | 99.99% | 99% | 99.90% | 99.95% | 99.99% |
| **ResNet-34** | | | | | | | | |
| Synflow | 86.03±0.71 | 61.61±10.76 | 56.33±3.44 | 10.00±0.00 | 57.44±1.72 | 22.91±18.98 | 1.00±0.00 | 1.00±0.00 |
| SET | 93.09±0.15 | 82.70±0.91 | 70.84±1.51 | 10.00±0.00 | **71.97±0.72** | 47.64±0.43 | 32.14±0.60 | 1.00±0.00 |
| RigL | 92.92±0.18 | 85.71±0.23 | 81.47±0.32 | 10.03±0.19 | 70.72±0.25 | 50.26±0.16 | 38.44±0.95 | 1.14±0.24 |
| EAST *(ours)* | **93.51±0.13** | **86.99±0.32** | **83.83±0.02** | **62.12±0.90** | 71.14±0.51 | **56.32 ±0.31** | **45.81±0.60** | **18.84±0.55** |
| DCTpS+RigL | 89.04±0.07 | 83.96±0.23 | 80.80±0.51 | 77.38±0.57 | 63.12±0.53 | 51.30±0.26 | 45.68±0.70 | 37.59±0.52 |
| DCTpS+RigL+EAST *(ours)* | **89.72±0.09** | **84.19±0.64** | **81.96±0.24** | **77.54±0.17** | **64.33±0.44** | **53.21±0.28** | **47.80±0.69** | **38.02±0.26** |
| **ResNet-50** | | | | | | | | |
| Synflow | 77.06±7.40 | 48.62±9.32 | 10.00±0.00 | 10.00±0.00 | 48.91±1.48 | 12.07±10.21 | 9.40±3.92 | 1.00±0.00 |
| SET | 91.97±0.25 | 27.05±15.17 | 16.04±2.85 | 10.00±0.00 | 70.35±0.11 | 8.59±2.61 | 1.26±0.28 | 1.00±0.00 |
| RigL | 92.75±0.25 | 58.41±42.03 | 30.14±10.85 | 10.00±0.00 | 70.50±0.32 | 51.90±0.62 | 7.02±5.33 | 1.00±0.00 |
| EAST *(ours)* | **93.48±0.13** | **86.16±0.12** | **81.18±0.59** | **49.55±4.69** | **71.04±0.15** | **56.76±0.22** | **45.55±0.43** | **3.64±2.84** |
| DCTpS+RigL | 89.66±0.27 | 85.98±0.55 | 85.03±0.25 | 83.13±0.53 | 64.85±0.23 | 55.98±0.09 | 52.71±0.20 | 49.63±1.11 |
| DCTpS+RigL+EAST *(ours)* | **89.67±0.12** | **87.10±0.35** | **85.87±0.39** | **83.88±0.19** | **64.92±1.41** | **58.17±0.22** | **55.60±0.38** | **51.33±0.57** |

Table 1: Accuracy comparison on CIFAR-10 and CIFAR-100 at different sparsity levels. Top section compares sparse training methods at each sparsity level. Bottom section shows results with DCTpS, which achieves higher accuracies at $\geq 99.95\%$ sparsity, but requires dense matrix computations. The best results for each category are in bold.

| Method | Inference FLOP (M) | | Inference | Network Size | | |
|---|---|---|---|---|---|---|
| | 99.95% | 99.99% | Comp. Cost | Theoretical | GPU-Supported | GPU-Supported Params |
| ResNet-50 (Dense) | 1297.83 | - | $\mathcal{O}(mn)$ | $\mathcal{O}(N)$ | $\mathcal{O}(N)$ | 23.5M |
| SET | 1.24 (0.001×) | 0.26 (0.0002×) | $\mathcal{O}(pmn)$ | $\mathcal{O}(PN)$ | $\mathcal{O}(N)$ | 23.5M |
| RigL | 1.24 (0.001×) | 0.26 (0.0002×) | $\mathcal{O}(pmn)$ | $\mathcal{O}(PN)$ | $\mathcal{O}(N)$ | 23.5M |
| EAST (w/o WS) | 1.24 (0.001×) | 0.26 (0.0002×) | $\mathcal{O}(pmn)$ | $\mathcal{O}(PN)$ | $\mathcal{O}(N)$ | 23.5M |
| EAST (w/ WS) | 2.59 (0.002×) | 0.52 (0.0004×) | $\mathcal{O}(pmn)$ | $\mathcal{O}(PN)$ | $\mathcal{O}(N - N_s)$ | 13.9M |
| DCTpS+RigL | 40.26 (0.031×) | 39.09 (0.030×) | $\mathcal{O}(q \log q + pmn)$ | $\mathcal{O}(PN)$ | $\mathcal{O}(N)$ | 23.5M |
| DCTpS+RigL+EAST | 40.26 (0.031×) | 39.09 (0.030×) | $\mathcal{O}(q \log q + pmn)$ | $\mathcal{O}(PN)$ | $\mathcal{O}(N)$ | 23.5M |

Table 2: Comparison of computational complexity and network size of ResNet-50 on different methods. FLOPs are given alongside multiplicative change from the dense model ($\times$). Let $N$ denote the total parameters in ResNet-50, $N_s$ the total shared parameters, $P \in (0, 1)$ the global density, and $m \times n$ the size of flattened weight tensors with density $p$, where $q = \max(m, n)$. "GPU-Supported Params" refers to the actual number of parameters on a commercial GPU without native support for irregular sparse patterns.

they are optimally configured in the paper, including $\Delta T$, ERK initialization, and the mask update interval of 1500 and 4000 for SET and RigL, respectively.

In addition, we benchmark DCT, the SOTA method specifically focused on extreme sparsity, using the repository provided by the method in Price & Tanner (2021). While this method performs very competitively at extreme sparsity settings, it comes at a cost due to the computation of a dense matrix matching the full architecture size, which adds an extra layer of computational overhead and requires extra memory. This method achieves the best performance when paired with RigL, which we evaluate, and then we combine it with our approach to evaluate if our method can improve it further. In this integration, we use only DyReLU phasing and cyclic density, leaving weight sharing to prevent conflicts with the DCT matrices embedded in the model architecture. Here, we use $\Delta T = 100$ to match results from the original paper, and we also use this value for our integration to ensure fairness.

For all DST methods, we use SGD with momentum as our optimizer. The momentum coefficient is set to 0.9, and L2 regularization coefficient is set to 0.0001. We set the training epoch to 250 and use a learning rate scheduler that decreases from 0.1 by a factor of 10X at halfway and three-quarters of way through training. For the DCT experiments, we train for 200 epochs with the Adam optimizer and a learning rate of 0.001. A more detailed description of the experimental setups and hyperparameters is included in the Supplementary Material. All experiments in our work are conducted on an A100 GPU.

**CIFAR-10 and CIFAR-100.** We evaluate EAST with ResNet-34 and ResNet-50 backbones, with a particular focus on extreme sparsities (99%-99.99%). We repeat our experiments three times and report the average accuracy in Table 1, which presents results across different configurations.

At 99%, all dynamic training methods achieve comparable performance, significantly outperforming Syn-Flow. This is unsurprising as a static mask is expected to lead to worse performance than a dynamic mask (Evci et al., 2021; Mostafa & Wang, 2019). EAST's advantages become increasingly pronounced at higher sparsities.

At 99.90% and 99.95%, EAST consistently outperforms existing methods across both architectures and datasets, with particularly notable improvements on ResNet-50 where competing methods experience severe accuracy degradation. While other approaches show dramatic performance collapse beyond 99.90% sparsity, EAST demonstrates more gradual degradation patterns.

At 99.99%, we observe that all other methods result in complete collapse to random performance across both datasets and architectures. In contrast, EAST maintains non-random accuracies. ResNet-50 shows greater sensitivity at this extreme sparsity, likely due to vanishing gradient issues exacerbated by the deeper architecture and reduced parameter density per layer.

Our evaluation of DCTpS reveals complementary strengths. While DCTpS excels at the highest sparsity levels (99.95% and 99.99%), classic DST methods including EAST maintain advantages at moderate extreme sparsities (99% and 99.9%) without requiring additional computational overhead. When combined with DCTpS, EAST consistently improves across all tested configurations, demonstrating the modular nature of our approach and its ability to enhance existing extreme sparsity methods.

**ImageNet.** Prior methods, such as SynFlow and DCTpS, were primarily evaluated on smaller datasets, such as CIFAR and Tiny-ImageNet. We extend these baselines to ImageNet by implementing and adapting their published code and evaluate the scalability of EAST on ImageNet using ResNet-50 at sparsity 99.5% and 99.9%. Given the much shortened training epochs for ImageNet, we implement only the DyReLU phasing and weight sharing components, excluding cyclic sparsity. As presented in Table 3, EAST has demonstrated scalability to large-scale datasets compared to existing methods. While SynFlow and SET collapse to random accuracies, EAST maintains stable performance across both tested sparsity levels and outperforms RigL by a notable margin.

Interestingly, we observe different dynamics on ImageNet compared to CIFAR experiments. While DCTpS showed competitive advantages on smaller datasets, it does not exhibit the same competitive advantages as seen before, even with the extra overhead. DCTpS alone results in performance collapse and it performs worse than RigL. When combined with RigL, the accuracy improvements are modest and may be attributed primarily to RigL's robust dynamic sparse exploration policy, rather than the DCT offset. On the inference efficiency front, EAST demonstrates practical advantages. Unlike DCTpS, which relies on a dense DCT matrix, EAST does not have such a drawback, as all DyReLU activations have been replaced with regular ReLU and thus runs just like the original ResNet model with ReLU. Therefore, EAST should theoretically have similar inference speed as the other DST methods, but faster than DCTpS for inference. To verify this, we ran both models for inference 100 times, taking their average and finding that DCTpS has $1.5\times$ higher latency and $1.33\times$ lower throughput than ours.

## 4.1 Complexity Analysis

The operations in neural networks can be framed in terms of matrix multiplication, allowing us to analyze the complexity of these methods. We report the FLOP following RigL's computation method in Table 2. If we examine sparse matrix with $pmn$ non-zeros, then theoretically the inference cost of EAST without weight sharing and other standard DST methods is just the cost of the sparse matrix, $\mathcal{O}(pmn)$, and their storage requirement is just $\mathcal{O}(PN)$. With weight sharing, EAST maintains the theoretical network size $\mathcal{O}(PN)$ while physically storing $\mathcal{O}(N - N_s)$ parameters, resulting in built-in compression on GPU's without native support for irregular sparsity patterns. It reduces the parameter count including zeros from 23.5M to 13.9M, but slightly increases the theoretical inference FLOP count due to the reuse of learnable parameters. This trade-off potentially offers practical benefits in memory-constrained devices.

| Method | 99.50% Sparsity | | 99.90% Sparsity | |
| --- | --- | --- | --- | --- |
| | Top-1 (%) | Top-5 (%) | Top-1 (%) | Top-5 (%) |
| SynFlow | 0.10 | 0.50 | 0.10 | 0.50 |
| SET | 0.10 | 0.50 | 0.10 | 0.50 |
| RigL | 32.50 | 58.58 | 8.53 | 24.18 |
| DCT | 0.10 | 0.50 | 0.10 | 0.50 |
| DCT+RigL | 19.99 | 43.34 | 6.62 | 18.84 |
| EAST *(ours)* | **34.45** | **62.44** | **11.23** | **29.47** |
| Inference Performance | | | | |
| | Throughput (imgs/sec) | | Latency (ms) | |
| DCT+RigL | 1229.25 | | 10.18 | |
| EAST *(ours)* | **1638.41** | | **6.80** | |

Table 3: Performance comparison on ImageNet at extreme sparsity levels using ResNet-50. For throughput and latency measurements, we use batch sizes of 2 and run 100 times and report the average.

| Model | Method | 99.95% Sparsity | 99.99% Sparsity |
| --- | --- | --- | --- |
| ResNet-34 | RigL w/ ReLU | 81.47 | 10.03 |
| | RigL w/ DyReLU | **81.80** | **54.58** |
| | RigL w/ WS | **81.96** | **62.94** |
| ResNet-50 | RigL w/ ReLU | 30.14 | 10.00 |
| | RigL w/ DyReLU | **69.57** | **53.32** |
| | RigL w/ WS | **82.02** | **62.71** |

Table 4: Ablating DyReLU phasing on DST with evaluation at different sparsities. All settings are kept the same except the use of DyReLU phasing vs standard ReLU. Experiments were run on CIFAR-10. Results show improved performance with DyReLU Phasing, especially at higher sparsity levels.

While DCTpS achieves higher accuracy than standard DST methods at extreme sparsity levels, it incurs substantial computational overhead, limiting its practical deployment benefits. DCTpS uses a dense DCT matrix, which theoretically could have a complexity of $\mathcal{O}(qlogq)$. We found that its theoretical inference FLOPs are about 32X higher than the standard DST at 99.95% and 150X higher at 99.99%. This is because the dense DCT computation dominates at extreme sparsities, becoming an increasingly significant bottleneck as sparsity increases. This additional computational overhead directly explains the inference throughput and latency differences observed in our experiments.

## 4.2 Ablation Studies

**Isolating DyReLU and Weight Sharing.** We first augment the standard RigL pipeline with only DyReLU phasing, replacing standard ReLU with DyReLU at initialization and gradually transitioning back to ReLU before the first learning rate schedule update. Weight sharing and cyclic sparsity are excluded in this isolated experiment. The results, summarized in Table 4 demonstrate a clear performance boost when utilizing DyReLU phasing, particularly at higher sparsity levels. At 99.99%, RigL completely loses its expressive capacity. However, as soon as DyReLU Phasing is used as a drop-in replacement for ReLU, it immediately starts converging. We observe that not only is training improved while DyReLU is in effect, but the performance continues to improve even after DyReLU has been completely phased out.

Improving gradient flow during early training has been shown to be critical for successfully training sparse networks. Evci et al. (2022) demonstrated that methods with better gradient flow, particularly during

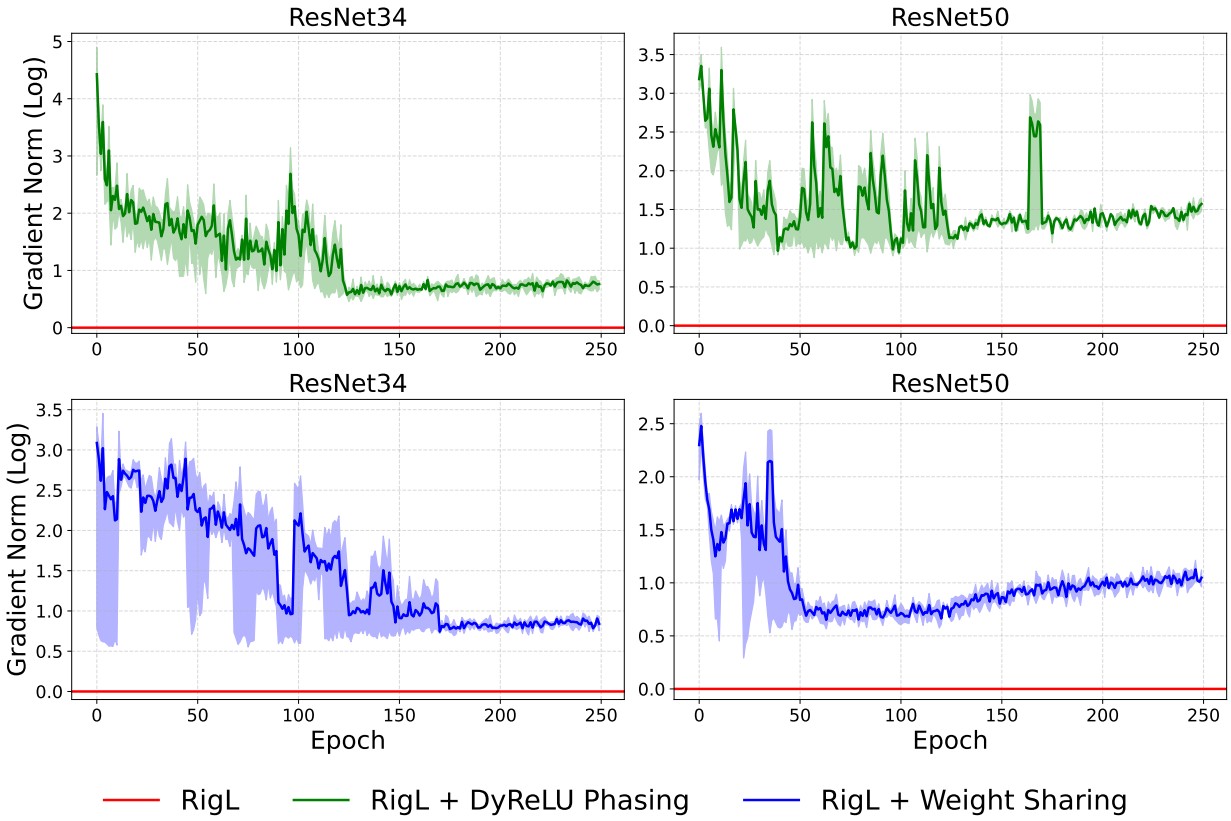

Figure 5: Gradient flow analysis. The top row compares gradient with and without DyReLU. The bottom row compares gradient with and without weight sharing.

initialization and early training, consistently achieved superior performance. Building on this insight, we investigated the effect of DyReLU phasing on gradient flow in addition to measuring accuracy.

To analyze this, we computed the sum of the gradient norm of all layers, and compared the gradient flow during training between models using DyReLU phasing against those with standard ReLU. As shown in the top row of Figure 5, models using only standard ReLU have zero gradient norms at 99.99% sparsity and this is reflected in their collapsed accuracy. On the other hand, the networks with DyReLU phasing receive high gradient flow in the beginning where $\beta = 1$, during phasing where $0 < \beta < 1$, and maintains healthy gradient flow when $\beta = 0$. Results suggest that the network retains some of the gradient-boosting benefits initially provided by DyReLU, We conjecture that this is because DyReLU amplifies important features in the beginning and helps the network settle into a configuration that preserves gradients even after switching to standard ReLU. Gradient flow especially during early stages is a strong indicator of the DST's success, and helps explain the performance gap between the methods.

Similarly, we isolate the effect of weight sharing by adding only this mechanism to the RigL pipeline, keeping all other settings the same. Once again, gradient flow has been restored, and we see a dramatic increase in gradient flow, as shown in the bottom row of Figure 5. This supports our hypothesis that creating additional gradient paths through weight sharing can help maintain stable training at extreme sparsities.

**Effect of Cyclic Sparsity.** We evaluate the impact of dynamically varying sparsity levels throughout training compared to maintaining static sparsity. Using ResNet-34 and ResNet-50 on CIFAR-100 at 99.95% target sparsity, we test different cyclic schedules against static sparsity baselines. As shown in Figure 6, cyclic sparsity consistently outperforms static sparsity across both architectures to varying degrees. The

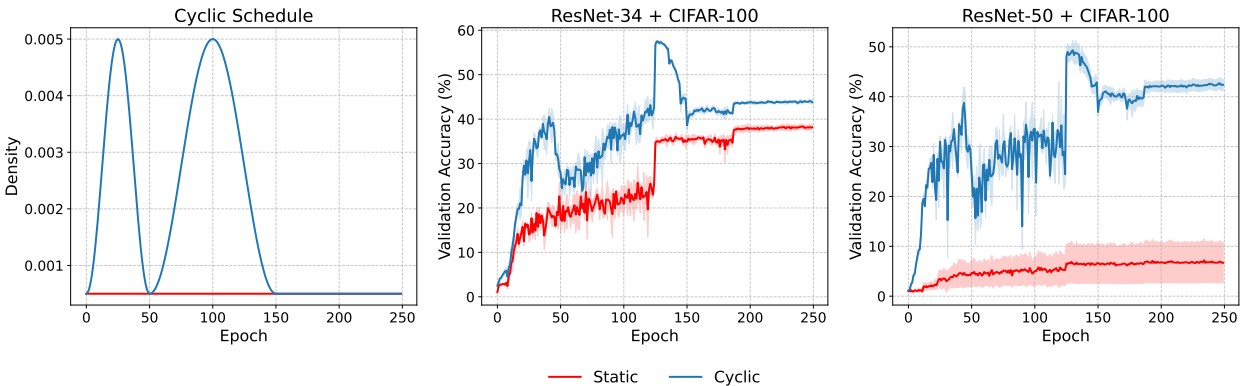

Figure 6: The cyclic sparsity schedule (left) and its effect on training a ResNet-34 (middle) and ResNet-50 (right) on CIFAR-100.

cyclic schedule allows the sparsity to temporarily relax before gradually returning to the target extreme level. This periodic increase in connectivity encourages parameter exploration during the denser phases while maintaining the desired final sparsity level. While ResNet-34 shows incremental improvement with cyclic scheduling, ResNet-50 demonstrates more substantial gains, suggesting that deeper networks may benefit more from this cyclic approach.

We provide more test results using different cyclic hyperparameters in Appendix C. Based on systematic evaluation across multiple configurations (Supplementary Tables C.2-C.5), we observe several trends. $10\times$ density multipliers generally outperform $3\times$, but $30\times$ is not necessarily better than $10\times$, suggesting there may be an elbow point where further increases no longer improves performance. Having two cycles generally performs better than having one, with longer subsequent cycle providing additional gains. While optimal configurations may vary by architecture and dataset, we recommend starting with a $10\times$ multiplier and a length multiplier of $2\times$ if two cycles are used when tuning cyclic sparsity hyperparameters.

## 5 Conclusion

In this paper, we introduced EAST, a new DST approach that focuses on extreme sparsity and incorporates DyReLU phasing, weight sharing, and cyclic sparsity scheduling. Through extensive empirical evaluation on CIFAR-10, CIFAR-10, and ImageNet using ResNet-34 and ResNet-50, we demonstrated that EAST achieves competitive performance at extreme sparsitys, surpassing existing methods by a large margin. Each component of EAST individually contributes to maintaining robust gradient flow and optimizing parameter exploration at extreme sparsity, while their combined effect can further enhance performance stability. Our results highlight the potential for practical applicability in resource-constrained environments, establishing it as a scalable and effective method for highly compressed networks. This work opens up future research to optimize models at extreme sparsities, an area often overlooked in previous studies.

**Acknowledgments**: Our work was supported by the UKRI EPSRC funding council (EP/V042270/1), a University of Aberdeen PhD studentship, the HPC facility "Maxwell", and the Interdisciplinary Institute.

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
