## A    Supplementary Material

## Section A: Preliminaries

### A.1 Dynamic Sparse Training

Let $\theta \in \mathbb{R}^N$ represent the set of all network weights, and let $\mathcal{M} \in \{0,1\}^N$ be a binary mask that defines the active connections in the sparse model:

$$\theta_{\mathcal{M}} = \theta \circ \mathcal{M}$$

where $\circ$ denotes element-wise multiplication. The sparsity $s$ of the network is the fraction of pruned weights, defined as:

$$s = 1 - \frac{\|\mathcal{M}\|_0}{\|\theta\|_0}$$

where $\|\mathcal{M}\|_0$ is the number of active (non-zero) weights in the mask $\mathcal{M}$ and $\|\theta\|_0$ is the total number of weights in a dense network. We follow the same criteria for updating connections used in RigL. First, we prune by magnitude, where the smallest magnitude weights are pruned. When pruning, the top-K operation selects the weights with the smallest absolute values:

$$\mathcal{M}_{\mathrm{prune}} = \mathrm{ArgTopK}(-|\theta_{\mathcal{M}}|, (1-s)\|\theta\|_0)$$

where $s$ is the sparsity level and $\mathcal{M}_{\mathrm{prune}}$ is the mask for pruned weights. We also adopt the gradient-based growth mechanism from RigL for regrowing weights, where the pruned weights with the largest gradients are reintroduced:

$$\mathcal{M}_{\mathrm{grow}} = \mathrm{ArgTopK}(|\nabla_\theta L|, s\|\theta\|_0)$$

where $\nabla_\theta L$ denotes the gradient of the loss function with respect to the weights. This ensures that important connections, as indicated by their gradients, are restored during training.

### A.2 Dynamic ReLU

The DyReLU activation introduces adaptive piecewise linear activation functions that adjust to the input. Given input tensor $x = \{x_c\}_{c=1}^C$, where $C$ is the number of channels, it is defined as:

$$y_c = f_{\theta(x)}(x_c) = \max_{1 \le k \le K} \{a_k^c(x)x_c + b_k^c(x)\}$$

where K is the number of functions, and k is the index. $a_k^c(x)$ and $b_k^c(x)$ are dynamically generated by a hyper-function $\theta(x)$ based on the input $x$. These dynamic coefficients allow DyReLU to provide more flexibility during training, and in our case enable better parameter exploration.

### B. Implementation Details

We share the hyperparameters used from Section 4 in Table B.1.

### C. Cyclic Sparsity Analysis

We tested two cyclic patterns - cosine schedules that follow smooth cosine curves and triangular schedules that transition linearly between top and bottom sparsity levels. The key hyperparameters are: the density multiplier (how many times the peak density exceeds the target sparsity - e.g., with a target density of 0.0005, a $3\times$ multiplier peaks at 0.0015 while $10\times$ peaks at 0.005), the number of cycles, and the cycle's length multiplier (how many times longer each subsequent cycle is compared to the previous one). As shown in Tables C.2-C.5, cyclic sparsity can be sensitive to hyperparameter choices and sometimes yields inconsistent results, particularly with ResNet-34 where certain configurations (e.g., cosine patterns with low multipliers) may decrease performance compared to static sparsity. This suggests that cyclic schedules may

| Model | Method | Data | BS | Epochs | LR | LR Drop Epochs | Opt | WD | Init | Cyclic | | | | $\Delta T_{dst}$ |
|---|---|---|---|---|---|---|---|---|---|---|---|---|---|---|
| | | | | | | | | | | n | l | m | $\Delta T_{cs}$ | |
| ResNet-34, EAST | | CIFAR-10/100 | 128 | 250 | 0.1 | 10x[75,150] | SGD(0.9) | 1.0e-4 | ERK | 2 | 50 | 10 | 350 | 4000 |
| ResNet-50 | DCT+RigL+EAST | CIFAR-10/100 | 128 | 200 | 0.001 | - | Adam | 5.0e-4 | ERK | 2 | 50 | 3 | 100 | 100 |
| ResNet-50 | EAST | ImageNet | 128 | 100 | 0.1 | 10x[25,50] | SGD(0.9) | 1.0e-4 | ERK | - | - | - | - | 4000 |

Table B.1: Experiment hyperparameters used in this paper. Batch Size (BS), Learning Rate (LR), Epochs, Learning Rate Drop (LR Drop), Optimizer with Momentum(Opt), Weight Decay (WD), Sparse Initialization (Init), Cyclic Sparsity-Related hyperparameters - Number of Cycles (n), Length of Cycle (l), Maximum Parameter Multiplier (m), Update Frequency during Cyclic Sparsity ($\Delta T_{cs}$), and Update Frequency for regular DST ($\Delta T_{dst}$).

require careful tuning to be effective. However, several consistent trends emerge across our experiments: (1) a higher density multiplier of $10\times$ generally outperform a lower one of $3\times$, (2) having two cycles tend to be more effective than just one, and (3) longer subsequent cycles often improve performance vs two cycles of equal length. While these patterns suggest that dynamic sparsity variation has merit, the high sensitivity to hyperparameters and occasional performance degradation indicate that cyclic sparsity should be considered a more experimental component of our framework, requiring further research to develop principled guidelines for hyperparameter selection.

| Method | Pattern | Multiplier | Cycles | Length Mult. | Test Accuracy (%) | Change (%) |
|---|---|---|---|---|---|---|
| Static | – | – | – | – | 81.47±0.47 | – |
| Cyclic | Cosine | 3 | 1 | - | 83.57±0.26 | +2.1 |
| Cyclic | Cosine | 3 | 2 | 1× | 83.98±0.14 | +2.5 |
| Cyclic | Cosine | 3 | 2 | 2× | 84.08±0.31 | +2.6 |
| Cyclic | Cosine | 10 | 1 | - | 84.27±0.13 | +2.8 |
| Cyclic | Cosine | 10 | 2 | 1× | 85.21±0.19 | +3.7 |
| Cyclic | Cosine | 10 | 2 | 2× | 85.34±0.18 | +3.9 |
| Cyclic | Cosine | 30 | 2 | 1× | 84.89±0.24 | +3.4 |
| Cyclic | Cosine | 30 | 2 | 2× | 84.94±0.39 | +3.5 |
| Cyclic | Triangular | 3 | 1 | - | 82.98±0.09 | +1.5 |
| Cyclic | Triangular | 3 | 2 | 1× | 84.16±0.06 | +2.7 |
| Cyclic | Triangular | 3 | 2 | 2× | 84.55±0.34 | +3.1 |
| Cyclic | Triangular | 10 | 1 | - | 84.32±0.15 | +2.8 |
| Cyclic | Triangular | 10 | 2 | 1× | 86.10±0.24 | +4.6 |
| Cyclic | Triangular | 10 | 2 | 2× | 85.46±0.04 | +4.0 |
| Cyclic | Triangular | 30 | 2 | 1× | 85.79±0.25 | +4.3 |
| Cyclic | Triangular | 30 | 2 | 2× | 85.51±0.14 | +4.0 |

Table 2: Cyclic sparse training results on CIFAR-10 with ResNet34

| Method | Pattern | Multiplier | Cycles | Length Mult. | Test Accuracy (%) | Change (%) |
|---|---|---|---|---|---|---|
| Static | – | – | – | – | 38.44±0.71 | – |
| Cyclic | Cosine | 3 | 1 | - | 34.37±0.23 | -4.1 |
| Cyclic | Cosine | 3 | 2 | 1× | 37.17±0.81 | -1.3 |
| Cyclic | Cosine | 3 | 2 | 2× | 38.36±0.48 | -0.1 |
| Cyclic | Cosine | 10 | 1 | - | 38.68±0.35 | +0.2 |
| Cyclic | Cosine | 10 | 2 | 1× | 41.35±0.54 | +2.9 |
| Cyclic | Cosine | 10 | 2 | 2× | 44.83±1.23 | +6.4 |
| Cyclic | Cosine | 30 | 2 | 1× | 42.45±1.04 | +4.0 |
| Cyclic | Cosine | 30 | 2 | 2× | 43.83±0.23 | +4.4 |
| Cyclic | Triangular | 3 | 1 | - | 34.63±1.52 | -3.8 |
| Cyclic | Triangular | 3 | 2 | 1× | 36.49±1.87 | -2.0 |
| Cyclic | Triangular | 3 | 2 | 2× | 38.80±1.10 | +0.4 |
| Cyclic | Triangular | 10 | 1 | - | 41.25±0.51 | +2.8 |
| Cyclic | Triangular | 10 | 2 | 1× | 43.97±0.74 | +5.5 |
| Cyclic | Triangular | 10 | 2 | 2× | 45.28±0.68 | +6.8 |
| Cyclic | Triangular | 30 | 2 | 1× | 44.15±0.24 | +5.7 |
| Cyclic | Triangular | 30 | 2 | 2× | 44.89±0.56 | +6.5 |

Table 3: Cyclic sparse training results on CIFAR-100 with ResNet34

| Method | Pattern | Multiplier | Cycles | Length Mult. | Test Accuracy (%) | Change (%) |
|---|---|---|---|---|---|---|
| Static | – | – | – | – | 30.15±10.85 | – |
| Cyclic | Cosine | 3 | 1 | - | 75.17±2.87 | +45.0 |
| Cyclic | Cosine | 3 | 2 | 1× | 75.67±3.19 | +45.5 |
| Cyclic | Cosine | 3 | 2 | 2× | 76.44±3.67 | +46.3 |
| Cyclic | Cosine | 10 | 1 | - | 78.75±4.06 | +48.6 |
| Cyclic | Cosine | 10 | 2 | 1× | 78.30±4.70 | +48.2 |
| Cyclic | Cosine | 10 | 2 | 2× | 77.37±4.87 | +47.2 |
| Cyclic | Cosine | 30 | 2 | 1× | 79.25±2.70 | +49.1 |
| Cyclic | Cosine | 30 | 2 | 2× | 79.68±1.79 | +49.5 |
| Cyclic | Triangular | 3 | 1 | - | 76.08±3.14 | +45.9 |
| Cyclic | Triangular | 3 | 2 | 1× | 75.06±4.40 | +44.9 |
| Cyclic | Triangular | 3 | 2 | 2× | 78.29±4.51 | +48.1 |
| Cyclic | Triangular | 10 | 1 | - | 78.55±5.29 | +48.4 |
| Cyclic | Triangular | 10 | 2 | 1× | 78.27±5.32 | +48.1 |
| Cyclic | Triangular | 10 | 2 | 2× | 79.23±4.93 | +49.1 |
| Cyclic | Triangular | 30 | 2 | 1× | 77.65±3.24 | +47.5 |
| Cyclic | Triangular | 30 | 2 | 2× | 79.01±4.32 | +48.9 |

Table 4: Cyclic sparse training results on CIFAR-10 with ResNet50

| Method | Pattern | Multiplier | Cycles | Length Mult. | Test Accuracy (%) | Change (%) |
|---|---|---|---|---|---|---|
| Static | – | – | – | – | 7.02±4.63 | – |
| Cyclic | Cosine | 3 | 1 | - | 33.71±1.57 | +26.7 |
| Cyclic | Cosine | 3 | 2 | 1× | 34.32±3.91 | +27.3 |
| Cyclic | Cosine | 3 | 2 | 2× | 35.34±3.95 | +28.3 |
| Cyclic | Cosine | 10 | 1 | - | 39.19±3.73 | +32.2 |
| Cyclic | Cosine | 10 | 2 | 1× | 39.61±2.83 | +32.6 |
| Cyclic | Cosine | 10 | 2 | 2× | 42.78±1.47 | +35.8 |
| Cyclic | Cosine | 30 | 3 | 1× | 36.68±2.95 | +29.7 |
| Cyclic | Cosine | 30 | 3 | 2× | 41.79±1.54 | +34.8 |
| Cyclic | Triangular | 3 | 1 | - | 36.19±2.20 | +29.2 |
| Cyclic | Triangular | 3 | 2 | 1× | 36.58±3.06 | +29.6 |
| Cyclic | Triangular | 3 | 2 | 2× | 37.61±3.21 | +30.6 |
| Cyclic | Triangular | 10 | 1 | - | 42.07±3.83 | +35.0 |
| Cyclic | Triangular | 10 | 2 | 1× | 43.55±1.57 | +36.5 |
| Cyclic | Triangular | 10 | 2 | 2× | 42.10±5.70 | +35.1 |
| Cyclic | Triangular | 30 | 2 | 1× | 41.95±2.57 | +34.9 |
| Cyclic | Triangular | 30 | 2 | 2× | 42.11±4.80 | +35.1 |

Table 5: Cyclic sparse training results on CIFAR-100 with ResNet50

## D. Training Cost Analysis

Table 6: Training Cost Analysis on CIFAR-100 with ResNet-34. Train time is measured on a single A100 GPU with batch size 128 for 250 epochs. Memory estimates are given by torchsummary using batch size of 1. "Param Size" reflects trainable parameters; "Est. Total" includes activations for a single forward/backward pass.

| Method | Train Time (hours) | Param Size (MB) | Est. Total (MB) |
|---|---|---|---|
| RigL | <1 | 81.36 | 100.44 |
| DCTpS | ∼2 | 162.66 | 197.36 |
| EAST w/ DyReLU | ∼1 | 81.36 | 107.81 |
| EAST w/ WS | <1 | 42.83 | 58.78 |
| EAST w/ Cyclic | <1 | 81.36 | 100.44 |
| EAST Combined | ∼1 | 54.96 | 78.78 |

EAST adds minimal training overhead compared to standard DST (RigL), while DCTpS requires 2× training time due to dense offset matrix computations.

- **DyReLU phasing:** Introduces additional parameters during early training (7 MB activation overhead), which are removed after phasing completes.

- **Weight sharing:** Directly reduces stored parameters by eliminating redundant weights across residual blocks (81→43 MB), providing immediate memory savings.

- **Cyclic sparsity:** Memory footprint identical to baseline RigL, as it only modifies the mask update schedule without additional parameters or activations.