# OpenReview forum: "Pushing the Limits of Sparsity: A Bag of Tricks for Extreme Pruning"
_TMLR — Accepted by TMLR_

### Review · Reviewer_DADF · 2025-08-25

**Summary Of Contributions:**

This paper investigates training sparse neural networks with extreme sparsity ratios, including 99.90%, 99.95%, and 99.99%. The paper introduces and explores three components to train sparse networks: 1) a new non-linearity called DyReLU that aims to preserve gradient flow, 2) a cyclic sparsity-level schedule where the amount of sparsity evolves during training, and 3) a weight-sharing component that reuses the same parameters across different layers.

The authors empirically explore various variants on the CIFAR-10, CIFAR-100, and ImageNet datasets, showing that their approach outperforms previous baselines such as RigL and SET in the high-sparsity regime (> 99.9%).

The authors also provide empirical ablations showing that 1) DyReLU and weight sharing improve gradient flow in the high-sparsity regime, and 2) the cyclic sparsity schedule outperforms a fixed sparsity schedule.

**Audience:**

Yes

**Broader Impact Concerns:**

No concern.

**Claims And Evidence:**

Yes

**Requested Changes:**

In addition to improving the manuscript's clarity (see weaknesses), I believe it is important to add an ablation study demonstrating the effect of each contribution (weight sharing, DyReLU, cyclic sparsity) on accuracy after pruning, across different datasets and sparsity ratios. Additionally, ensuring that the baselines use properly tuned hyperparameters for the high-sparsity regime would help validate the correctness of the empirical setup.

**Strengths And Weaknesses:**

Strengths:
- Authors propose an extensive empirical section that shows their approach is competitive with previous work.
- Authors validate the effect of weight-sharing/DyReLU on potential gradient flows


Weaknesses:
- Clarity of the paper could be improved:
1) some crucial descriptions regarding contribution are DyReLU in appendix A. This content should probably be moved in the main paper as it's one of the main paper contributions.
2) Even after reading appendix A, some details regarding DyReLU are still not clear. How do you define the 'hyper-function' to compute a^c_k(x) and b^c_k(x)? 3) I am not sure I understand line 3-5 in algorithm 1? Should it be an if-statement instead of a for line 3 and what is \phi line 4?

- It's not clear what is the impact of each of the three proposed components. For instance DCTpS+EAST does not use weight-sharing yet achieves the best performance at sparsity level, which raises the question of importance of weight-sharing. On the other hand, ImageNet experiment does not use cyclic sparsity. Overall, it would be nice to ablate the effect of the different components (DyReLU, cyclic sparsity, weight-sharing) in the different setup and for different sparsity level.

- Authors use previous hyperparameters for the different baseline but it is not clear that those hyperparameters are optimal in their setups (i.e. high-sparsity ratio). Is the empirical setup similar to those of previous work? If not, what is the effect of hyperparameter tuning of the baseline?

---

> ### Author Response · Authors · 2025-11-05
> **Response to reviewer DADF**
>
> >In addition to improving the manuscript's clarity (see weaknesses), I believe it is important to add an ablation study demonstrating the effect of each contribution (weight sharing, DyReLU, cyclic sparsity) on accuracy after pruning, across different datasets and sparsity ratios. Additionally, ensuring that the baselines use properly tuned hyperparameters for the high-sparsity regime would help validate the correctness of the empirical setup.
>
> We appreciate the feedback. We have significantly improved clarity throughout the paper.
>
> We have moved the DyReLU phasing description from the Supplementary Materials to Section 3.1. The phasing schedule performs continuous interpolation between DyReLU and ReLU coefficients using a decay factor β that linearly decreases from 1 (pure DyReLU) to 0 (pure ReLU) over epochs $T_s$ to $T_e$. We have also corrected Algorithm 1 line 3 from 'For' to 'If to properly reflect the conditional logic consistent with our implementation. In the algorithm, $\phi$ represents the activation function applied - we initialize with DyReLU and gradually transition to standard ReLU during the phasing period through coefficient interpolation.
>
> We provide comprehensive component ablations in Section 4.2, where Table 2 and Figure 5 isolate the effects of weight sharing and DyReLU phasing. Figure 6 shows the effect of cyclic sparsity, and Supplementary Tables C2-C5 further analyze cyclic sparsity hyperparameters. Our ablation strategy validates two questions: 1) Do individual components provide benefits? The tables and figures in Section 4.2 demonstrates each component improves over RigL baseline. 2) Does the combined method generalize? Table 1 and 3 show EAST Combined achieves consistent improvements across multiple models, dataset, sparsities tested.Running every component combination at every sparsity on every dataset for multiple seeds would provide diminishing returns. Our approach efficiently validate component effectiveness through focused ablations while demonstrating generalization through comprehensive evaluation with the combined method.
>
> Regarding the hyperparameters, as described in Section 4, we adopt configurations directly from ITOP, which specifically studied optimal update hyperparameters in DST methods. We use their reported settings across all baselines and our method to isolate the effect of our contributions and ensure maximum fairness in comparison.

---

### Review · Reviewer_QjVn · 2025-10-03

**Summary Of Contributions:**

This paper proposes EAST (Extreme Adaptive Sparse Training), combining three techniques—DyReLU phasing, weight sharing, and cyclic sparsity—to train neural networks at extreme sparsity levels (99.90%-99.99%). While the work addresses an interesting problem, the paper suffers from significant methodological limitations, outdated literature review, insufficient experimental scope, and missing practical considerations that undermine its contributions.

**Audience:**

Yes

**Broader Impact Concerns:**

No particular concers

**Claims And Evidence:**

Yes

**Requested Changes:**

1. Update the related work section to include recent literature from 2023-2024, covering advances in sparse training, modern pruning techniques, hardware-aware sparsity methods, and recent activation function research. The current literature review is unacceptably outdated for a submission to a 2025 venue.

2. Provide concrete evidence and citations supporting the claim of "increasing demand" for extreme sparsity.

3. Include comprehensive training cost analysis comparing EAST against all baselines in terms of wall-clock training time, memory usage, and training FLOPs.

4. Expand experimental evaluation to include modern architectures, particularly Vision Transformers (ViT, DeiT, Swin) and recent CNNs (EfficientNet, ConvNeXt). Evaluate on larger-scale and more diverse datasets beyond CIFAR.  The current scope (ResNets on CIFAR) is too narrow to support generalizability claims.

5. Provide principled guidance for setting cyclic sparsity hyperparameters rather than acknowledging the difficulty and leaving it to future work

6. Compare EAST against alternative compression strategies including structured pruning, quantization, and knowledge distillation to smaller dense models.

7. Analyze actual memory footprint and inference characteristics on real hardware (not just theoretical FLOPs) to substantiate deployment claims.

**Strengths And Weaknesses:**

## Strengths

1. The paper tackles the challenging problem of maintaining network performance at extreme sparsity levels (99.90%-99.99%), which represents a significant technical challenge even if the practical motivation is unclear. The proposed EAST framework demonstrates that non-random accuracy can be maintained at these extreme sparsities where existing methods completely collapse, showing clear improvements over baselines like RigL and SET.

2. The gradient flow analysis in Figure 5 provides valuable insights into why DyReLU phasing and weight sharing improve training stability. The systematic ablation studies isolate the contribution of each component, demonstrating that DyReLU phasing alone can restore gradient flow and prevent collapse at 99.99% sparsity. The experimental methodology is generally sound, with multiple runs and error bars reported.

3. The modular nature of the approach is noteworthy, as demonstrated by successfully integrating EAST components with DCTpS to achieve consistent improvements. This suggests the techniques could potentially be adapted to other sparse training frameworks.

## Weaknesses

1. The related work section is severely outdated, relying almost entirely on papers from 2018-2021 with virtually no citations from 2023-2024. Given the rapid pace of development in sparse training, pruning, and efficient neural networks, this is a critical flaw. It is hard to address the impact of the work within the current state of the field when that "current state" is represented by 3-4 year old papers. Recent advances in hardware-aware sparsity, modern dynamic sparse training methods, and contemporary activation function research are completely absent.

2. The central motivation for pursuing extreme sparsity lacks any supporting evidence. The authors claim there is "increasing demand" to push toward extreme sparsities in "resource-constrained settings," yet provide no citations, no real-world applications, no deployment scenarios, and no evidence that accuracy trade-offs of 30+ percentage points would ever be acceptable. At 99.99% sparsity, EAST achieves 62% on CIFAR-10 compared to 93% at 99% sparsity—a catastrophic accuracy loss for marginal additional compression. The paper reads as solving a problem that may not meaningfully exist beyond academic curiosity.

3. The central motivation for pursuing extreme sparsity lacks any supporting evidence. The authors claim there is "increasing demand" to push toward extreme sparsities in "resource-constrained settings," yet provide no citations, no real-world applications, and no deployment scenarios; additionally,  no evidence that accuracy trade-offs would ever be acceptable in a real-world scenario. At 99.99% sparsity, EAST achieves 62% on CIFAR-10 compared to 93% at 99% sparsity, which looks like a catastrophic accuracy loss for marginal additional compression.

4. The complete absence of training cost analysis is inexcusable. The paper introduces multiple complex mechanisms: DyReLU adds computational overhead during phasing, cyclic sparsity requires dynamic mask updates, and weight sharing introduces gradient accumulation complexity. Yet there is no comparison of training time, memory usage, or training FLOPs versus baselines. The authors dismiss this by stating their "primary focus is on the deployable model," but this omission fundamentally limits the paper's contribution.

5. The experimental scope is far too limited. Testing only ResNet-34 and ResNet-50 ignores the reality that Vision Transformers have become standard in computer vision, and modern CNN architectures like EfficientNet and ConvNeXt are widely used. The reliance on CIFAR-10/100 (32×32 toy datasets) for primary results is insufficient by modern standards. While ImageNet is included, the results are limited and reveal concerning inconsistencies, contradicting the paper's narrative. No evaluation on more difficult datasets, or any domain-specific datasets where extreme compression might actually matter, undermines claims of practical relevance.

6. The paper acknowledges that cyclic sparsity has a "large number of possible configurations makes systematic experimentation and hyperparameter tuning challenging," but provides minimal guidance or a principled approach to setting these hyperparameters. The weight sharing mechanism increases inference FLOPs while reducing parameters, yet there is no analysis of actual memory savings on real hardware or discussion of how shared parameters affect convergence properties. Table 2 shows that "GPU-Supported Params" remains O(N) for most methods, suggesting the practical deployment advantage is limited even at extreme sparsities where commercial accelerators typically support 90-98% sparsity efficiently anyway.

7. The paper lacks comparison with alternative compression strategies, including structured pruning, quantization (which often achieves better compression-accuracy trade-offs), or knowledge distillation to smaller dense models. Without these comparisons, it is impossible to assess whether extreme unstructured sparsity is even the right approach.

---

> ### Author Response · Authors · 2025-11-05
> **Response to reviewer QjVn**
>
> We highly appreciate the reviewer's detailed and constructive feedback. We have carefully addressed all the points and revised the paper, as detailed below.
>
> >Update the related work section to include recent literature from 2023-2024, covering advances in sparse training, modern pruning techniques, hardware-aware sparsity methods, and recent activation function research. The current literature review is unacceptably outdated for a submission to a 2025 venue.
>
> We appreciate this and fully agree. We have updated the related work section to incorporate recent development in sparse training from 2024-2025, including SRigl, DynaDiag, DSCR, CHT. While these represent different problem settings, they provide important context for the broader sparse training lanscape.
>
>
> >Provide concrete evidence and citations supporting the claim of "increasing demand" for extreme sparsity.
>
> >The paper reads as solving a problem that may not meaningfully exist beyond academic curiosity
>
> We appreciate this feedback and we have rephrased our claim in the introduction to better reflect our scientific motivation. The original intent of this work was primarily curiosity-driven - we observed that sparse network accuracy collapsed abruptly beyond certain thresholds and sought to explore the fundamental limits of sparse training. We have reframed the second paragraph of our introduction to position this as scientific exploration (ie., what are the limits of sparse training) rather than claiming immediate practical demand for 99%+ compression. While edge deployment motivates studying extreme sparsity, our main contribution is showing it is actually possible and understanding what makes it work. As with many advances in machine learning, such fundamental research can inform future practical application once feasibility is demonstrated.
>
> >Include comprehensive training cost analysis comparing EAST against all baselines in terms of wall-clock training time, memory usage, and training FLOPs.
>
> Our study was scoped around the limit of compressibility (“how small we can squeeze a CNN while remaining functional”), not training efficiency ("how much work it takes to squeeze"). In real world, training cost is a separate matter because models are not trained on devices they are deployed on. We have clarified this scope in the introduction.
>
> Regarding training costs, wall-clock training time for all base DST methods (SET, RigL etc) are equivalent to dense training on standard hardware without specialized sparse kernels. This fundamental limitation is why literatures involving unstructured sparsity typically report FLOP and sparsity instead of wall-clock measurements. EAST adds modest overhead only before DyReLU is fully replaced with ReLU, and after that the training time matches the baseline methods. Without shifting the paper's focus, we added a training cost analysis table in Supplemental Materials Section D summarizing training overhead for each component.
>
> - DyReLU phasing: Adds compute proportional to the DyReLU auxiliary parameters during $T_s \rightarrow T_e$; after phasing (about halfway through training in our runs), the model reverts to standard ReLU and the cost matches standard DST.
>
> - Weight sharing: Results in lower peak parameter memory (shared tensors stored once) but higher forward/backward FLOPs because shared weights participate in multiple blocks. This trade-off is reflected in Table 2.
>
> - Cyclic sparsity: Overhead stems from (i) occasional prune/regrow top-k steps and (ii) temporarily higher density during the grow phase. The cost scales with the schedule amplitude and update frequency.

---

> ### Author Response · Authors · 2025-11-05
> **Response to QjVn**
>
> >Expand experimental evaluation to include modern architectures, particularly Vision Transformers (ViT, DeiT, Swin) and recent CNNs (EfficientNet, ConvNeXt). Evaluate on larger-scale and more diverse datasets beyond CIFAR. The current scope (ResNets on CIFAR) is too narrow to support generalizability claims.
>
> We appreciate the suggestion to expand to other architectures. CNN is the scope of this study for the following reasons:
>
> - Our weight-sharing mechanism is explicitly designed to exploit the repetitive residual blocks where weights are re-used across blocks (Figure 4). While ViT architectures do contain repetitive blocks that could potentially support weight-sharing, it would require addressing different questions - which components to share (Q/K/V projection? FFN layers?), and how to handle different layer dimensions, etc. This requires a substantial architectural redesign rather than a straightforward extension. Similarly, ViT uses GeLU or other activations instead of ReLU, and it would require separate investigations of which activations are suitable.
>
> - The foundational DST methods we used to build upon (RigL, SET, ITOP, DCTpS) were developed and evaluated exclusively on CNNs. Our work maintains consistency with this established scope. Additionally, we would need to adapt these methods to ViT first to establish baselines. This would require substantial modifications, making fair comparison challenging without established reference implementations.
>
> We evaluate on CIFAR-10, CIFAR-100 and imagenet, the standard benchmarks used by all baseline DST methods. The extreme sparsity regime we target has not been explored on additional datasets in prior DST literature, making it difficult to establish meaningful comparison beyond standard benchmarks.
>
> >Provide principled guidance for setting cyclic sparsity hyperparameters rather than acknowledging the difficulty and leaving it to future work
>
> We ran additional experiments involving additional cyclic hyperparameters. We have revised Section 4.2 and provided principled guidance for setting cyclic sparsity hyperparameters based on observations from Supplementary Tables C2-C5.
>
> >Compare EAST against alternative compression strategies including structured pruning, quantization, and knowledge distillation to smaller dense models.
>
> We focus on structured dynamic sparse training, which is complementary to quantization, knowledge distillation and structured pruning. These techniques are not competitive to each other but can be used in combination, ie., pruning + quantization, pruning + knowledge distillation, unstructured + structured pruning are all well-established pipelines. None of the fundamental DST methods we build upon compared against these alternative paradigms. Additionally, there is no established baselines we can find for these methods at extreme sparsities on our experimental setup. It would require extensive implementation from scractch beyond this work's scope.
>
> >Analyze actual memory footprint and inference characteristics on real hardware (not just theoretical FLOPs) to substantiate deployment claims.
>
> Real hardware measurements for unstructured sparsity require specialized hardware support, which is why unstructured pruning and DST literatures use theoretical FLOPs as standard practice. We do provide actual inference measurements where meaningful - Table 3 shows our method achieves 1.5x lower latency and 1.33x higher throughput on A100, and this difference comes from avoiding DCTpS's dense matrix overhead, not from sparsity itself. The network size (GPU-supported params) in Table 2 also reflects actual on-device memeory measurements. Weight-sharing directly removes repetitive residual blocks, providing immediate memory savings even without specialized sparse kernels. The newly added section D in Supplemental Materials should also reflect how each component may behave on real hardware. However, please keep in mind that values measured in Supplementary Table 6 were measured without specialized sparse kernel so it does not take sparsity into account. We observe that weight sharing immediately removes memory usage while including DyReLU temprorily increases it during training, as expected.

---

### Review · Reviewer_QHqX · 2025-10-26

**Summary Of Contributions:**

The paper introduces a collection of complementary techniques designed to stabilize and improve **dynamic sparse training (DST)** at extreme sparsity levels (e.g. >99.9%), where prior methods collapse. The authors present **EAST (Extreme Adaptive Sparse Training)**, a modular framework combining three main innovations:
1. **Dynamic ReLU (DyReLU) Phasing** - replacing standard ReLU with Dynamic ReLU early in training to enhance gradient flow and parameter exploration, then linearly phasing it out to standard ReLU before the first learning-rate decay. This preserves gradient pathways while removing DyReLU parameters for inference.

2. **Weight Sharing (WS)** - intra-layer parameter reuse across residual blocks in ResNet architectures, multiplying each learnable parameter’s participation in forward and backward paths while maintaining identical learnable parameter count. This improves gradient propagation and utilization without increasing model size.

3. **Cyclic Sparsity Scheduling** - varying the network’s sparsity level cyclically between minimum and maximum densities during training, alternating between dense and sparse phases. This encourages parameter exploration and prevents layer collapse before converging to the final target sparsity.

They integrate these techniques within standard DST frameworks (e.g., **RigL, SET**) and achieves consistent accuracy improvements at extreme sparsities on CIFAR-10/100 and ImageNet, using ResNet-34/50 backbones. The framework also complements **DCTpS** by improving its accuracy without additional inference cost. Ablation studies demonstrate the benefits of DyReLU, WS, and cyclic sparsity on gradient flow, measured through gradient norm analysis and positive pre-activation statistics.

**Audience:**

Yes

**Broader Impact Concerns:**

There are no explicit ethical concerns beyond those common to general-purpose model-compression research.

**Claims And Evidence:**

Yes

**Requested Changes:**

I think the including following items would make the paper stronger:
1. Provide stronger theoretical or empirical justification for different components of the proposed method -- e.g. for why DyReLU phasing preserves gradients post-transition. In this case, a controlled analysis (e.g., gradient flow visualization or distributional change before and after phasing) would substantiate this mechanism.
2. Some clarifications about weight-sharing -- how is the accumulation of gradients on shared tensors done, i.e. are the gradients are averaged, summed, or rescaled?
3. **IMPORTANT** -- I would really like to see some experiments on non-vision or non-ResNet architectures.

**Strengths And Weaknesses:**

# Strengths
1. **Good integration of complementary mechanisms**: The paper shows how three orthogonal strategies - dynamic ReLU, structural parameter reuse, and sparsity scheduling - can be effectively combined, each addressing distinct failure modes in extreme sparse regimes.
2. **Consistent improvements over baselines methods**: Experiments across multiple architectures (ResNet-34/50) and datasets (CIFAR-10/100, ImageNet) with multiple baselines (SynFlow, SET, RigL, DCTpS) show consistent improvements, particularly beyond 99.9% sparsity.
3. **Extensive ablations**: Each component of EAST is analyzed in isolation and in combination. The supplementary material includes detailed hyperparameters and cyclic sparsity sensitivity analysis, increasing reproducibility.
4. **Practical efficiency**: Unlike DCTpS, EAST avoids dense-matrix computations during inference. FLOP and latency comparisons highlight usefulness for real-world deployability.

# Weaknesses
1. **Lack of theoretical grounding**: The paper is almost entirely empirical. The mechanisms (especially DyReLU phasing and cyclic sparsity) are motivated heuristically, without formal analysis of why or when they should succeed or fail.
2. **Limited exploration of general architectures**: All experiments are on image classification with ResNets. It is unclear if EAST generalizes to **transformers** or NLP models, where sparse training dynamics differ. This would be very useful to have.

---

> ### Author Response · Authors · 2025-11-05
> **Response to reviewer QHqX**
>
> We thank the reviewer for the constructive feedback. We have carefully addressed the issues and revised the paper, as detailed below.
>
> >Provide stronger theoretical or empirical justification for different components of the proposed method -- e.g. for why DyReLU phasing preserves gradients post-transition. In this case, a controlled analysis (e.g., gradient flow visualization or distributional change before and after phasing) would substantiate this mechanism.
>
> We provide empirical justification through gradient flow analysis and positive pre-activation tracking in Figure 3. Figure 5 in Ablation Studies visualizes gradient flow throughout training and demonstrates that DyReLU phasing maintains healthy gradient norms even after complete conversion to ReLU (visible gradient flow vs 0 for ReLU-only). We have enhanced Section 3.1 to better connect these empirical findings to the underlying mechanism.
>
> >Some clarifications about weight-sharing -- how is the accumulation of gradients on shared tensors done, i.e. are the gradients are averaged, summed, or rescaled?
>
> Gradients on shared tensors are summed by PyTorch's autograd. When a shared weight tensor participates in multiple forward passes, backpropagation accumulates gradients from all usage sites. Each shared block multiplies the shared weights by learnable scaling factors, which receive independent gradients and allow blocks to specialize despite using the same base weights. We have revised Section 3.2 to clarify this.
>
> >IMPORTANT -- I would really like to see some experiments on non-vision or non-ResNet architectures.
>
> We appreciate the suggestion to expand to other architectures. CNN is the scope of this study for the following reasons:
>
> - Our weight-sharing mechanism is explicitly designed to exploit the repetitive residual blocks where weights are re-used across blocks (Figure 4). While ViT architectures do contain repetitive blocks that could potentially support weight-sharing, it would require addressing different questions - which components to share (Q/K/V projection? FFN layers?), and how to handle different layer dimensions, etc. This requires a substantial architectural redesign rather than a straightforward extension. Similarly, ViT uses GeLU or other activations instead of ReLU, and it would require separate investigations of which activations are suitable.
>
> - The foundational DST methods we used to build upon (RigL, SET, ITOP, DCTpS) were developed and evaluated exclusively on CNNs. Our work maintains consistency with this established scope. Additionally, we would need to adapt these methods to ViT first to establish baselines. This would require substantial modifications, making fair comparison challenging without established reference implementations."

---

### Decision · Action_Editor_WFr6 · 2025-11-28

**Recommendation:** Accept as is

**Additional Comments:**

The reviewers highlighted that the contribution is primarily empirical, that the experimental scope is restricted to image classification benchmarks, and that the current method does not yet demonstrate clear FLOP or latency gains on standard hardware. Nevertheless, the proposed Extreme Adaptive Sparse Training method presents a thorough and well-executed study of an underexplored extreme sparsity regime, offering practical insights and tools for training sparse convolutional networks at these levels.

PS: in the abstract, please fix “ImageNet,achieving” -> “ImageNet, achieving”.

**Audience:**

Yes

**Audience Explanation:**

The topic is within TMLR’s scope on efficient deep learning and sparse training. There is a clear scientific interest for researchers working on pruning, dynamic sparse training, model compression, and training dynamics. The paper demonstrates that standard dynamic sparse training methods fail catastrophically beyond 99.9% sparsity, and that a carefully designed combination of mechanisms can maintain model functionality in this regime.

All three reviewers agree that the work is interesting and relevant to the TMLR audience, even though the empirical scope is limited to convolutional neural networks.

**Claims And Evidence:**

Yes

**Claims Explanation:**

The paper presents an extensive empirical evaluation of the proposed Extreme Adaptive Sparse Training (EAST) method on ResNet-34 and ResNet-50 across CIFAR-10, CIFAR-100, and ImageNet, with comparisons to multiple established sparse training baselines, including RigL, SET, DCTpS, and SynFlow. All three reviewers agree that the core empirical claims are supported.

The revised version strengthens the evidence by clarifying the description and implementation of the components, adding ablations that isolate DyReLU phasing, cyclic sparsity, and weight sharing, and expanding empirical analyses of gradient flow and practical aspects such as training cost and hardware considerations in the supplementary material.